# The Influence of Using a Footstool during a Prolonged Standing Task on Low Back Pain in Office Workers

**DOI:** 10.3390/ijerph16081405

**Published:** 2019-04-18

**Authors:** Michelle D. Smith, Chun Shing Johnson Kwan, Sally Zhang, Jason Wheeler, Tennille Sewell, Venerina Johnston

**Affiliations:** 1School of Health and Rehabilitation Sciences, The University of Queensland, Brisbane 4072, Australia; chun.kwan1@uqconnect.edu.au (C.S.J.K.); sallykim30@gmail.com (S.Z.); j.wheeler@uqconnect.edu.au (J.W.); t.ricketts@uq.net.au (T.S.); v.johnston@uq.edu.au (V.J.); 2Recover Injury Research Centre, The University of Queensland, Brisbane 4006, Australia

**Keywords:** musculoskeletal symptoms, low back pain, standing workstation, footstool, office workers

## Abstract

While many office workers experience low back pain (LBP), little is known about the effect of prolonged standing on LBP symptoms. This repeated-measures within-subjects study aimed to determine whether office workers with LBP are able to work at a standing workstation for one hour without exacerbating symptoms and whether using a footstool affects LBP severity. Sixteen office workers with LBP performed computer work at a standing workstation for one hour under the following two conditions, one week apart: with a footstool and without a footstool. The intensity of LBP was recorded at 10 min intervals. Maximal severity of LBP pain and change in LBP severity throughout the standing task were not different between the footstool and no footstool conditions (*p* > 0.26). There was a trend for more participants to have an increase in their pain between the start and end of the task when not using a footstool compared to using a footstool (*p* = 0.10). Most office workers with LBP are able to use a standing workstation without significant exacerbation of symptoms, but a proportion will experience a clinical meaningful increase in symptoms. Using a footstool does not change the severity of LBP experienced when using a standing workstation in individuals with a history of LBP.

## 1. Introduction

Prolonged sitting has been associated with negative health outcomes which is a concern for computer-based workers who typically sit for over six hours per day [1]. Standing workstations offer the opportunity for posture variation [2]; however, prolonged standing can also have negative implications, particularly the development of pain or musculoskeletal conditions affecting the legs and lumbar spine [3]. Recent laboratory studies have identified that standing for 30–60 min resulted in the development of transient low back pain (LBP) in 40% of people without a history of LBP [4]. Another study has reported that those who develop transient LBP with prolonged standing are three times more likely to need medical care for LBP within three years [5]. Having regular sitting breaks may help relieve the progression of LBP, but the utility of this strategy and the duration of sitting breaks needed are unclear [4].

In order to appropriately manage LBP, it is important to consider the factors that affect its development. While research findings are often inconclusive [6,7,8], the following risk factors for LBP have been suggested: increased lumbar lordosis [6,7], a lack of movement [9], and reduced sagittal plane postural variability [10]. Laboratory-based studies have shown that the development of LBP during standing can be significantly reduced by standing on a sloped surface [11,12] or performing full trunk flexion at 15 min intervals [13]. Standing on a sloped surface was associated with decreased co-activation of the gluteus minimus muscle bilaterally, which may influence LBP development [12]. Intermittently flexing the spine is thought to decrease LBP by moving the spine out of a compressive lordotic posture and relaxing (decreasing the activation of) the lumbar erector spinae and gluteus maximus muscles [13]. An alternative means of encouraging trunk flexion during standing is to place a foot on an elevated platform [6]. As increased lumbar lordosis is proposed to be a risk factor for the development of LBP during standing and intermittent trunk flexion has been associated with decreased standing-induced LBP, using a footstool may help to reduce LBP in office workers who are using standing workstations. This may also be particularly important for office workers with current LBP. Little is known about the effect of prolonged standing or the use of a footstool in office workers with pre-existing LBP, as most of the current literature includes pain-free populations. In light of the high prevalence of LBP in office workers [14] and LBP being a leading cause of years lived with disability [15], we require a better understanding of how to manage LBP in office workers.

The aim of this study was to determine: (i) whether office workers with LBP are able to work at a standing workstation for one hour without exacerbating symptoms; and (ii) whether using a footstool affects the severity of LBP experienced when using a standing workstation in office workers with LBP.

## 2. Material and Methods

A within-subject repeated-measures study design was used to compare the severity of LBP experienced during a prolonged standing task with and without a footstool. The one-hour testing sessions were conducted in a private room in a laboratory at the participants’ workplace.

### 2.1. Participants

Volunteers for this study were recruited through online newsletters, posters, and electronic notice boards on a University campus. Participants were eligible for inclusion if they were 18 years old or older, performed sitting computer work for more than 30 h per week, and had non-specific LBP for ≥3 months that was rated in severity as ≥2/10 on an 11-point numeric rating scale (NRS) anchored with “no pain” at 0 and “worst pain imaginable” at 10. Exclusion criteria were: current pregnancy, trauma or surgery to the back or legs in the last 12 months, dizziness or loss of balance when standing, or inability to stand for one hour. All participants provided informed consent prior to participation in the study. The study was conducted in accordance with the Declaration of Helsinki, and the protocol was approved by the Ethics Committee of The University of Queensland (#2015000469).

### 2.2. Prolonged Standing Task 

Participants completed a prolonged standing task [16] while performing their usual computer-based work duties for an hour under two different conditions: with a footstool (elevated platform) and without a footstool. Participants stood within a rectangular space (122 cm × 60 cm) with their body fist-width in front of a height-adjustable workstation. The workstation was standardised to each participant so that the desk height was 5–6 cm below the lateral epicondyle, the computer monitor was an arm’s length from the body, and the top of the computer monitor was at eye level. During the footstool condition, the height of the footstool was adjusted so that the hip was flexed to 45° when the foot was resting on top of the footstool [17]. Participants were instructed to weight-shift as much as they liked but to refrain from leaning on the desk. For the footstool condition, participants were also instructed to use the footstool as much as they liked.

A body map and 11-point NRS were used to record the location, type, and intensity of symptoms experienced throughout the standing task. This information was recorded at the start of the standing task, and at 10 min intervals throughout the 60 min test. The investigator verbally asked the participants to rate their pain on an NRS. If the participants rated their pain >0 (indicating the presence of pain), they were asked to mark the location of pain on a paper body chart that was placed in front of them. The NRS used to assess symptoms intensity is a reliable and valid outcome measure for LBP [18] with excellent test–retest reliability [19]. The 11-point NRS has been used to record LBP at 10–15 min intervals in numerous previous studies using a prolonged standing task [16,20,21,22]. Participants also provided their own free-form descriptors of the type of symptoms experienced. Participants were reminded at the 10 min intervals that they could shift their weight as often as desired. During the footstool condition, participants were also reminded to place either foot on the footstool as often as desired. Researchers recorded all changes in standing posture (including asymmetrical weight-bearing and use of the footstool) throughout the task. Participants were permitted to stop the test at any time. The test was also stopped if participants experienced pain >7/10 on the NRS.

### 2.3. Experimental Protocol

The order of test conditions was randomised by flipping a coin. The two testing sessions were conducted at the same time of day, approximately one week apart. Participants were asked to wear the same shoes for each assessment, and the workstation set-up (i.e., height of the desk and monitor) was standardized between the footstool and no footstool conditions. At the end of the second session, participants were asked whether they preferred using the footstool, not using the footstool, or if they had no preference.

In order to describe the study sample, the following data were collected using an online survey completed at the start of the first testing session: age, height, weight, severity of LBP over the previous 7 days, and level of disability. The Oswestry Disability Index was used to record the impact of LBP on functional activities [23]. It contains 10 items that are summed to create an overall score ranging from 0 to 50, with higher scores indicating greater disability.

### 2.4. Data and Statistical Analysis

The change in maximal severity of LBP (the difference between the maximal pain experienced during the test and that experienced at baseline) and the change in pain severity from baseline to the end of the prolonged standing task were compared between conditions (footstool and no footstool). As data were not normally distributed, Wilcoxon Paired Signed Rank tests were used to compare changes in pain severity between conditions, and data are presented as median and inter-quartile range (IQR). The number of participants who experienced (i) a reduction of pain (lower reported NRS LBP severity at the completion compared to the start of the prolonged standing task), (ii) no change in pain (no difference in the reported NRS LBP severity at the start and end of the prolonged standing task), or (iii) worsening of pain (higher reported NRS LBP severity at the completion compared to the start of the prolonged standing task) were compared using a chi-square test. The number (percentage) of participants who reported their pain to be 2 points higher or lower on the NRS at the end of the standing tasks (which suggests a clinically meaningful change in pain [23]) are reported. Participant preference (those who preferred standing with the footstool, without the footstool, or had no preference), number of times participants used the footstool, and total time (in minutes) of footstool use during the prolonged standing task are reported descriptively. Statistical analysis was performed using Statistica 64 (Dell Software), with significance set at *p* < 0.05.

## 3. Results

Sixteen participants (eight males and eight females) with mean (SD) age, weight, height, and body mass index of 40 (12.3) years, 73.1 (13.6) kg, 1.72 (0.10) m, and 24.7 (3.0) kg/m^2^ respectively, participated in this study. Mean (SD) worst pain in the last seven days was 3.9 (1.9) out of 10 on the NRS, and the Oswestry score was 5.1 (4.4) out of 50. During the footstool and no footstool conditions, 75% (n = 12) and 88% (n = 14) of participants experienced LBP during the prolonged standing task. The trajectory of mean pain experienced at each time point throughout in the prolonged standing task during both the footstool and no footstool conditions is shown in Figure 1. The most common self-selected pain descriptors used by participants during both the footstool and the no footstool conditions were discomfort (50%, n = 8 vs. 31%, n = 5), stiffness (38%, n = 6 vs. 44%, n = 7), and ache (19%, n = 3 vs. 25%, n = 4), with many participants using more than one description of their symptoms. All participants completed testing, and the test was not stopped for any participant because of pain.

There was no significant difference in the change in maximal severity of LBP pain experienced throughout the prolonged standing task between the footstool (median (IQR): 1 (0; 1.25)) and no footstool (median (IQR): 1 (0; 2)) conditions (*p* = 0.68). The change in the severity of LBP experienced by participants from the start to the end of the 60 min standing task was also similar between conditions (median (IQR) footstool: 0 (−0.13; 0.25), no footstool: 1 (0; 2); *p* = 0.14). 

The proportion (number) of participants who experienced worsening of their pain, improvement in their pain, and no change in their pain between the start and the end of the prolonged standing task when using the footstool and not using the footstool, and the mean change in pain severity are reported in Table 1. Although not statistically significant, there was a trend for more participants to have an increase in their pain between the start and the end of the task when not using a footstool compared to using a footstool (*p* = 0.10; Table 1). Throughout the standing task, 31% (5 out of 16) of participants in the no footstool condition and 19% (3 out of 16) of participants in the footstool condition reported a worsening of pain of ≥2 out of 10 on the NRS, which suggests a clinically meaningful change in pain (*p* = 0.74). Two out of 16 participants (13%) reported an improvement in pain of ≥2 points on the NRS for each of the conditions. 

There was considerable variation in the use of the footstool when it was available (range: 3–32 times used; 1–54 min of use). Throughout the 60 min standing test, the mean (SD) number of times the footstool was used was 14.9 (9.1), and the mean (SD) total time that a foot was resting on the footstool was 26.0 (15.2) min. Half (50%, n = 8) of participants preferred working at the standing workstation when using a footstool, 25% (n = 4) preferred the no footstool condition, and 25% (n = 4) had no preference.

## 4. Discussion

This study aimed to determine whether office workers with LBP are able to work at a standing workstation for one hour without adversely effecting symptoms, and to determine whether using a footstool affects the severity of LBP experienced when office workers with LBP work at a standing workstation. Our study found that most people with chronic LBP were able to stand for one hour while performing their usual computer-based work duties without a significant increase in their LBP. However, a proportion of individuals experienced a clinically significant increase in their pain (of >2 out of 10 on an NRS [24]) when performing work a standing workstation, irrespective of footstool use. Our data indicated a trend for more study participants to experience worsening of their LBP during the task when not using a footstool compared to when using a footstool. This may suggest that while using a footstool in conjunction with a standing workstation may not decrease the severity of LBP experienced, it may help to prevent an increase in low back symptoms in some office workers with a history of LBP. However, it must also be noted that an equal number of participants (n = 4) reported worsening and improvement in their LBP when using the footstool, which suggests considerable variability between individuals. 

While pain varied among participants, the change in pain through the one-hour prolonged standing task in both conditions was small and did not exceed the clinical important difference for most individuals. This suggests that most individuals with LBP can work at a standing workstation for one hour without clinically significant increases in pain. However, a proportion of individuals did experience exacerbations in their pain when using a standing workstation both with and without a footstool. This needs to be considered when using standing workstations, and office workers with a history of LBP should be educated to pay close attention to their pain and to change positions accordingly. 

Our findings that there are no differences in the severity of LBP experienced when office workers with a history of LBP stand with and without a footstool are similar to those from Lee et al who investigated the effect of using a footstool on standing-induced LBP in asymptomatic individuals [25]. Authors found no difference in LBP severity when using a footstool and not using a footstool during a two-hour standing task. Our data adds to these findings, showing that the severity of LBP is unaffected by footstool use in individuals with a history of LBP as well as in those with no history of LBP. Our data indicate that using a footstool may help to prevent an increase in low back symptoms in some office workers with a history of LBP; however, this was not statistically different between conditions. 

The variability in pain when using and not using the footstool among the study participants suggests that there may be factors associated with positive (improvements in pain) and negative (worsening of pain) responses to standing with and without a footstool in individuals with a history of LBP. Using a footstool may reduce lumbar lordosis during standing [6]. As increased lumbar lordosis has been suggested to be a risk factor for LBP during standing [6,7], a footstool may be beneficial to individuals who stand in a lordotic posture. Use of a footstool may address other proposed risk factors, such as lack of movement [8] and reduced sagittal plane postural variability [10]. Moving the foot onto and off of a footstool may facilitate increased movement, and having the foot positioned on or off of the footstool may add variability to sagittal plane posture. However, alternatively, it is possible that not using a footstool may facilitate a greater variability in movement, as individuals may shift their weight differently when movement is not perceived to be limited to lifting a leg on and off a footstool. Because of the diversity of LBP presentations and suggestion of LBP sub-groups with different aggravating factors and postures [26], it may not be surprising that individuals respond differently to using and not using a footstool. Further research is needed to identify factors that are associated with positive and negative changes in symptoms when office workers are working at a standing workstation and the relationship of such factors with footstool use. 

The protocol used in this study is based on that used in previous publications and was designed to replicate an office working environment by allowing participants to complete their own computer-based work during the prolonged standing task and self-select their use of the footstool during the footstool condition. However, there are limitations that also must be considered. First, participant self-selection of timing and duration of footstool use meant that there was considerable variability in use between study participants. While participants were reminded to use the footstool as much as they liked and the self-selection replicates real-world use, low footstool use in some individuals may have influenced the comparisons between the footstool and no footstool conditions. Second, the small sample size (n = 16), low level of baseline pain, and high level of functioning (mean Oswestry score of 5.1/50) of our study participants may require replication of this study with larger samples and in individuals with higher baseline pain levels and lower functioning to confirm the generalisability of our findings. Third, the prolonged standing task in this study was one hour, whereas many other studies have used a two-hour standing task. This duration was chosen as the majority of healthy participants in our previous study developed LBP within the first hour of the standing task [20], and guidelines for standing at work do not recommend people stand for more than one hour at a time [27]. 

## 5. Conclusions

This study identified that most office workers with a history of LBP are able to use a standing workstation without significant exacerbation of symptoms; however, a proportion of individuals experience a clinically significant increase in pain, irrespective of using or not using a footstool. Footstool use does not change the severity of LBP experienced during a prolonged standing task, but it may decrease the chance of some individuals experiencing an increase in LBP during standing. These data suggest that office workers with LBP who are using standing workstations should carefully monitor symptoms, and the use of a footstool should be based on an individual’s pain response and preference. Future research should look to identify characteristics of individuals who do and do not respond positively to using standing workstations and footstools, so that recommendations and assessments of standing workstation and footstool suitability can be developed.

## Figures and Tables

**Figure 1 ijerph-16-01405-f001:**
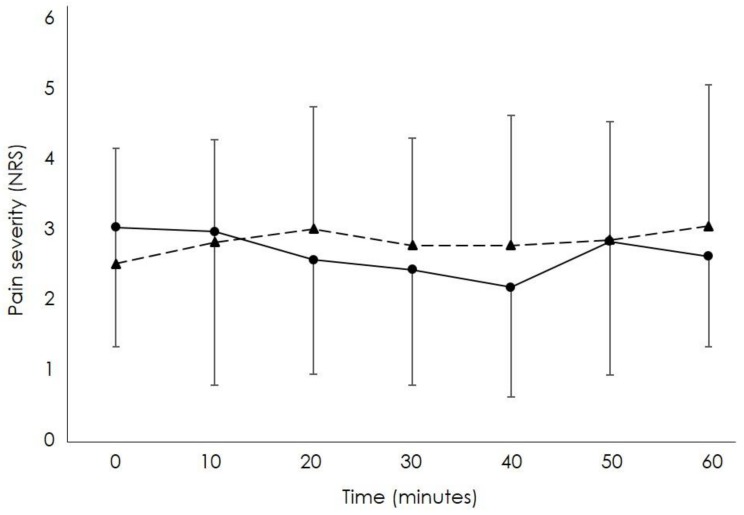
Mean severity of low back pain (LBP) rated on a numerical rating scale (0 = “no pain”; 10 = “worst pain imaginable”) throughout the 60 min prolonged standing task when using a footstool (solid line; circles) and not using a footstool (dashed line; triangles).

**Table 1 ijerph-16-01405-t001:** Proportion of participants who experienced improvement, worsening, or no change in low back pain and mean (standard deviation (SD)) change in pain between the start and end of the prolonged standing task.

Change in Low Back Pain	Footstool Condition		No Footstool Condition	
	% (n)	Mean (SD)	% (n)	Mean (SD)
Improvement	25% (n = 4)	2.7 (1.7)	13% (n = 2)	3.5 (2.1)
Worsening	25% (n = 4)	2.6 (2.0)	63% (n = 10)	1.9 (1.7)
No change	50% (n = 8)	0 (0)	25% (n = 4)	0 (0)

% = percentage; n = number.

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
