# Peer review of "The Influence of Using a Footstool during a Prolonged Standing Task on Low Back Pain in Office Workers"

_ijerph, 2019, doi:10.3390/ijerph16081405_

Round 1
Reviewer 1 Report
This is a generally well-written article investigating changes in LBP symptoms during a standing task, and whether the use of a footstall can change such symptoms. Whilst the research question is of value, there are currently areas of the paper that need addressing before being suitable for publication. In particular, there seem to be an imbalance in the literature review and discussion, with a focus on the probable benefits of footstall use for LBP patients. The results do not conclusively support this finding, and this should be reflected in the write up. Specific comments Title: Given that there is no calculation of ‘effect size’ the title of the paper could be altered to more accurately reflect its contents. Abstract: Lines 14-16 Please make it clear in this section that the single group performed the task both with a footstall and without with a 1 week separation. As a reader I was unsure as to whether this was a comparison between two different groups or the same group repeated. Introduction: Lines 35-36 The end of this sentence could be re-written, as it currently it not clear Lines 38-39 and 47 Whilst there is some evidence for the degree of lordosis being related to LBP, the majority of the literature to date is more inconclusive. This should be reflected in the critique of the literature at this stage. Line 51 Office worker(s) Line 52 Remove the word ‘the’ after [13]? Given that previous studies have shown that ‘non LBP’ participants can also develop LBP during standing tasks, why did you not choose to include such a control group? Methods Lines 62-64 Is there any more detail you could provide regarding the LBP participants? E.g. were there participants with radicular symptoms, or was the group made up of non-specific LBP participants? Line 94 and Line 123 Given that over three quarters of participants in both conditions experienced LBP during the exercise, and this was expected, did you consider how this may this have affected the second task? Using the methodology described (i.e. testing 1 week apart) it would seem likely that there would be some sort of behavioural change in the second task due to fear avoidance etc. Why therefore did you decide to repeat the task, and not compare two separate groups? Results Line 145 remove ‘or’ replace with ‘of’ Line 152 insert the word ‘of’ after n=8 Discussion Paragraph 1 and Table 1 Given that there were an equal number of participants who improved or worsened under the footstall condition, it is felt that this should be reflected more in the discussion. Currently the discussion focusses very much on the likely positive impact of the footstall, but the results do not support this unidirectional standpoint. Lines 185-187 It is also shown that the use of a footstall may worsen LBP symptoms in this group. Lines 188-200 You mention the above at the start of this paragraph, but then describe only the likely positive impact mechanisms of using a footstall. It may be suitable to also highlight how the opposite may be the case and why? It may also be of value to discuss possible sub-groups of LBP patients that may respond differently during such tasks (i.e. O’Sullivan’s flexion/extension aggravation groups). Line 184 Given that previous studies of 2 hours duration revealed similar results, do you feel that a 1 hour task is long enough to change symptoms? This could be commented on in the discussion. Lines 208-211 These are indeed the key limitations, and they could be expanded upon. Given that the pain severity rating was around 2-3 at baseline, then for any clinically meaningful change to happen, there would need to be almost complete pain resolution in most cases, which is perhaps unrealistic. Future studies may best recruit participants with higher baseline scores. Conclusions Please address the conclusions to provide a more balanced reflection of the study’s results as per the comments above.Author Response
Dr Miya Zhang
Assistant Editor
IJERPH
11 April 2019
Dear Dr Zhang,
Re: Manuscript ijerph-478526: The effect of using a footstool during a prolonged standing task on low back pain in office workers
Thank you very much for giving us the opportunity to revise our submitted manuscript. We have addressed the comments from Reviewer #1 on a point by point basis in the table below and made amendments to the manuscript accordingly.
Thank you again and looking forward to hearing from you in due course.
Yours sincerely
Michelle Smith
Response to comments from Reviewer #1
Reviewer’s comment | Author response | |
1 | Whilst the research question is of value, there are currently areas of the paper that need addressing before being suitable for publication. In particular, there seem to be an imbalance in the literature review and discussion, with a focus on the probable benefits of footstall use for LBP patients. The results do not conclusively support this finding, and this should be reflected in the write up. | The introduction and discussion have been edited to reflect the reviewer’s concern about an imbalance in the literature review. The reviewer’s specific comments with regards to this have been address below. |
2 | Given that there is no calculation of ‘effect size’ the title of the paper could be altered to more accurately reflect its contents. | The word “effect” has been removed from the title of the paper. The title now reads “The influence of using a footstool during a prolonged standing task on low back pain in office workers”.
|
3 | Abstract: Lines 14-16 Please make it clear in this section that the single group performed the task both with a footstall and without with a 1 week separation. As a reader I was unsure as to whether this was a comparison between two different groups or the same group repeated. | The abstract has been edited to more clearly indicate that one group of participants performed the tasks one week apart. The added text is underlined below.
Revised text: “This repeated measures within subjects study aimed to determine whether office workers with LBP are able to work at a standing workstation for one hour without exacerbating symptoms, and whether using a footstool affects the LBP severity. Sixteen office workers with LBP performed computer work at a standing workstation for one hour under the following two conditions one week apart: with a footstool and without a footstool.”
|
4 | Introduction: Lines 35-36 The end of this sentence could be re-written, as it currently it not clear. | This sentence has been re-written to read as follows: “Having regular sitting breaks may help relieve the progression of LBP, but the utility of this strategy and duration of sitting breaks needed are unclear [4].”
|
7 | Introduction: Lines 38-39 and 47 Whilst there is some evidence for the degree of lordosis being related to LBP, the majority of the literature to date is more inconclusive. This should be reflected in the critique of the literature at this stage. | The sentence at lines 38-39 has been rewritten to acknowledge inconsistencies in the literature. It now reads as: “While research findings are often inconclusive [6,7], the following risk factors for LBP have been suggested: increased lumbar lordosis [6,7], a lack of movement [8], and reduced sagittal plane postural variability [9].”
The following reference has been added to support this statement: Sadler SG, Spink MJ, Ho A, De Jonge XJ, Chuter VH. Restriction in lateral bending range of motion, lumbar lordosis, and hamstring flexibility predicts the development of low back pain: a systematic review of prospective cohort studies. BMC Musculoskelet Disord. 2017;18:179. https://doi.org/10.1186/s12891-017-1534-0.
The sentence at line 47 has been edit to reflect that lumbar lordosis has been proposed to be a risk factor for LBP. The sentence now reads: “As increased lumbar lordosis is proposed to be a risk factor for the development of LBP during standing and intermittent trunk flexion has been associated with decreased standing-induced LBP, using a footstool may help to reduce LBP in office workers who are using standing workstations.”
|
8 | Introduction: Line 51 Office worker(s) Line 52 Remove the word ‘the’ after [13]? | This word has been removed. |
Introduction: Given that previous studies have shown that ‘non LBP’ participants can also develop LBP during standing tasks, why did you not choose to include such a control group? | The majority of research investigating standing workstations in office worker has been undertaken in pain-free populations. In light of this, we specifically wanted to investigate the use of standing workstation (with and without the addition of the footstool) in officer workers with LBP. Further, previous research has investigated the use of a footstool in individuals without LBP [21].
| |
9 | Methods: Lines 62-64 Is there any more detail you could provide regarding the LBP participants? E.g. were there participants with radicular symptoms, or was the group made up of non-specific LBP participants? | Study participants had non-specific LBP. This has been added to the eligibility criteria.
The edited text is underlined in the following sentence: “Participants were eligible for inclusion if they were 18 years old or older, performed sitting computer work for more than 30 hours per week, and had non-specific LBP for ³3 months that was rated in severity as ³2/10 on an 11-point numeric rating scale (NRS) anchored with "no pain" at 0 and "worst pain imaginable" at 10.”
|
10 | Methods: Line 94 and Line 123 Given that over three quarters of participants in both conditions experienced LBP during the exercise, and this was expected, did you consider how this may this have affected the second task? Using the methodology described (i.e. testing 1 week apart) it would seem likely that there would be some sort of behavioural change in the second task due to fear avoidance etc. Why therefore did you decide to repeat the task, and not compare two separate groups? | We choose to use a repeated measures within subject design for this study to enable us to compare using and not using a footstool within one participants. This designed replicated that used previously to assess footstool use in asymptomatic individuals [21] and accounted for likely differences in standing-induced LBP between participants. To account for possible effects of testing order between the footstool and no footstool conditions, order was randomised for each participant.
The following sentence was added to the start of the methods to explain the study design: “A within subject repeated measures study design was used to compared severity of LBP experienced during a prolonged standing tasks with and without a footstool. »
|
11 | Results: Line 145 remove ‘or’ replace with ‘of’. Line 152 insert the word ‘of’ after n=8 | I believe the change to line 145 has been made by the editor as there is no “or” in this sentence.
The word “of” has been inserted in line 152 as recommended.
|
12 | Discussion: Paragraph 1 and Table 1 Given that there were an equal number of participants who improved or worsened under the footstool condition, it is felt that this should be reflected more in the discussion. Currently the discussion focusses very much on the likely positive impact of the footstall, but the results do not support this unidirectional standpoint. | Text has been added to the first paragraph of the discussion to emphasise the variability in responses between participants and state that equal numbers of participants reported worsening and improvement in LBP symptoms when using the footstool.
|
13 | Discussion: Lines 185-187 It is also shown that the use of a footstall may worsen LBP symptoms in this group. | Text has been added to the previous paragraph to further emphasise that a proportion of individuals using and not using a footstool experienced worsening of their LBP symptoms. The text was added to this paragraph as it was thought to link more closely with the content of this paragraph than the following paragraph, and it allowed a balance presentation of worsening of symptoms in some participants in both the footstool and no footstool conditions.
The underlined text has been added: “However, it must be noted that a proportion of individuals did experience exacerbations in their pain when using a standing workstation both with and without a footstool.”
|
14 | Discussion: Lines 188-200 You mention the above at the start of this paragraph, but then describe only the likely positive impact mechanisms of using a footstall. It may be suitable to also highlight how the opposite may be the case and why? It may also be of value to discuss possible sub-groups of LBP patients that may respond differently during such tasks (i.e. O’Sullivan’s flexion/extension aggravation groups). | Text has been added to this paragraph to provide suggestions for the positive impact of not using a footstool, and also to identify that sub-groups of individuals with LBP may respond differently to standing with and without a footstool.
Added text: “However, alternatively, it is possible that not using a footstool may facilitate greater variability in movement as individuals may shift their weight differently when not perceived to be limited by lifting a leg on and off a footstool. Due to the diversity of LBP presentations and suggestion of LBP sub-groups with different aggravating factors and postures, it may not be surprising that individuals respond differently to using and not using a footstool.”
The following reference has been added to support the suggestion of sub-groups: Dankaerts W, O'Sullivan P, Burnett A, Straker L. Altered patterns of superficial trunk muscle activation during sitting in nonspecific chronic low back pain patients: importance of subclassification. Spine (Phila Pa 1976). 2006;31:2017-23. https://doi.org/10.1097/01.brs.0000228728.11076.82.
The final sentence of this paragraph (which has not been altered) indicates the need for further research in this area.
|
15 | Discussion: Line 184 Given that previous studies of 2 hours duration revealed similar results, do you feel that a 1 hour task is long enough to change symptoms? This could be commented on in the discussion. | The limitation of using a 1-hour standing task has been identified and the selection of this duration has been justified in the limitations section of the discussion.
Added text: “Third, the prolonged standing task in this study was 1-hour; whereas, many other studies have used a 2-hour standing task. This duration was chosen as the majority healthy participants in our previous study developed LBP within the first hour of the standing task, and guidelines for standing at work do not recommend people stand for more than one hour at a time.”
The following references have been added to support this justification: Johnston V, Gane EM, Brown W, et al. Feasibility and impact of sit-stand workstations with and without exercise in office workers at risk of low back pain: A pilot comparative effectiveness trial. Appl Ergon. 2019;76:82-89. https://doi.org/10.1016/j.apergo.2018.12.006. Waters, T. R., and R. B. Dick. 2015. "Evidence of Health Risks Associated with Prolonged Standing at Work and Intervention Effectiveness." Rehabilitation Nursing 40 (3). doi: 10.1002/rnj.166. |
16 | Discussion: Lines 208-211 These are indeed the key limitations, and they could be expanded upon. Given that the pain severity rating was around 2-3 at baseline, then for any clinically meaningful change to happen, there would need to be almost complete pain resolution in most cases, which is perhaps unrealistic. Future studies may best recruit participants with higher baseline scores. | The limitation of low baseline pain levels in our study has been emphasised in the limitations and we have suggested that future studies recruit individuals with higher baseline pain levels.
The edited text is underlined in the following sentence in the discussion: “Second, the small sample size (n=16), low level of baseline pain and high level of functioning (mean Oswestry score of 5.1/50) of our study participants may require replication of this study with larger samples and individuals with higher baseline pain levels and lower functioning to confirm the generalisability of our findings.”
|
17 | Conclusions: Please address the conclusions to provide a more balanced reflection of the study’s results as per the comments above. | We have added text to the conclusion to indicate that some individuals experience worsening of their LBP in both the footstool and no footstool conditions.
The edited text is underlined in the following sentence: “This study identified that most office workers with a history of LBP are able to use a standing workstation without significant exacerbation of symptoms; however, a proportion of individuals experience a clinical significant increase in pain irrespective of using or not using a footstool.”
|
Reviewer 2 Report
This paper compares the development of pain in office workers with mild backpain during working at a standing workplace for one hour with or without using a footstool. Generally, this manuscript is well written. However, there are two main concerns:
Firstly, how valid is the NRS in measuring the intensity of spontaneous back pain at intervals of 10 minutes? The provided reference 17 reported reliability of NRS for measurement intervals of 1 and 4 weeks. Reference 18 stated that single NRS values recorded at hourly intervals show high variation between consecutive days, but their average value was stable and reproducible. The pain provoked by prolonged standing might be more reliable described by subtracting the mean of the measurements at 10 to 60 minutes from baseline score.
How was this pain recording organised? Did the study instructor verbally remind the participants every 10 minutes to score their back pain? Was a computer programme used for that purpose? Was an acoustic signal used to indicate that the next back pain assessment is due? Who recorded the pain score: the participants in writing on paper or their computer or the study instructor? How did pain scoring interfere with their computer work?
Secondly, it remains unclear where the study was conducted? On site, where the participants were employed as office workers? In a laboratory of the University? How many of the participants were self-employed? If the study was conducted in the workers office, did it interfere with their working duties, was it scheduled at the beginning, the middle or the end of their working day? Did the participants perform their routine computer work? Were the computer tasks compatible with interruptions every 10 minutes?
If conducted at the university, how long were participants off duty on the date of the experiment? All this information is essential for understanding, whether the use of a standing workstation is a valid alternative in the real working life of patients with mild back pain.
Minor points of criticism are:
Student´s t-test requires a normal distribution of data. Was this checked?
Whilst pain worsening is clearly defined as increase of NRS score equal or greater than 2 points, a definition of improvement (= pain reduction?) is missing.
The body mass index should be added to anthropometric characteristics of participants.
After major revision acceptance of the publication is recommended.
Author Response
Dr Miya Zhang
Assistant Editor
IJERPH
11 April 2019
Dear Dr Zhang,
Re: Manuscript ijerph-478526: The effect of using a footstool during a prolonged standing task on low back pain in office workers
Thank you very much for giving us the opportunity to revise our submitted manuscript. We have addressed the comments from Reviewer #2 on a point by point basis in the table below and made amendments to the manuscript accordingly.
Thank you again and looking forward to hearing from you in due course.
Yours sincerely
Michelle Smith
Response to comments from Reviewer #2
Reviewer’s comment | Author response | |
1 | This paper compares the development of pain in office workers with mild backpain during working at a standing workplace for one hour with or without using a footstool. Generally, this manuscript is well written. However, there are two main concerns: Firstly, how valid is the NRS in measuring the intensity of spontaneous back pain at intervals of 10 minutes? The provided reference 17 reported reliability of NRS for measurement intervals of 1 and 4 weeks. Reference 18 stated that single NRS values recorded at hourly intervals show high variation between consecutive days, but their average value was stable and reproducible. The pain provoked by prolonged standing might be more reliable described by subtracting the mean of the measurements at 10 to 60 minutes from baseline score. | The NRS has been used to record changes in pain at 10-15 minute intervals during a prolonged standing tasks in a number of previous studies. The following sentence has been added to the methods to indicate this: “The 11-point NRS has been used to record LBP at 10-15 minute intervals in numerous previous studies using a prolonged standing task.”
And the following references to support this statement have been added: 1. Sorensen, C.J.; Johnson, M.B.; Callaghan, J.P.; George, S.Z.; Van Dillen, L.R. Validity of a Paradigm for Low Back Pain Symptom Development During Prolonged Standing, The Clinical journal of pain 2015, 31, 652-659. 2. Nelson-Wong, E., Gregory, D.E., Winter, D.A., Callaghan, J.P., 2008. Gluteus medius muscle activation patterns as a predictor of low back pain during standing. Clin. Biomech. 23, 545–553. 3. Nelson-Wong, E., Callaghan, J.P., 2010a. Changes in muscle activation patterns and subjective low back pain ratings during prolonged standing in response to an exercise intervention. J. Electromyogr. Kinesiol. 20, 1125–1133. 4. Johnston V, Gane EM, Brown W, et al. Feasibility and impact of sit-stand workstations with and without exercise in office workers at risk of low back pain: A pilot comparative effectiveness trial. Appl Ergon. 2019; 76:82-89.
|
2 | How was this pain recording organised? Did the study instructor verbally remind the participants every 10 minutes to score their back pain? Was a computer programme used for that purpose? Was an acoustic signal used to indicate that the next back pain assessment is due? Who recorded the pain score: the participants in writing on paper or their computer or the study instructor? How did pain scoring interfere with their computer work? | Pain recording was undertaken by a study investigator on a paper data collection sheet. The investigator verbally asked the participant to rate their pain in the NRS. If the indicated the presence of any pain (by stating a number greater than 0), a paper body chart that was placed on the desk in front of them for them to mark their location of pain. The body chart was removed as soon as it was completed so it did not interfere with the participants work.
The following text has been added to the methods to explain this: “The investigator verbally asked the participant to rate their pain on a NRS. If the participate rated their pain >0 (indicating the presence of pain), they were asked to mark the location of pain on a paper body chart that was placed in front of them.”
|
3 | Secondly, it remains unclear where the study was conducted? On site, where the participants were employed as office workers? In a laboratory of the University? How many of the participants were self-employed? If the study was conducted in the workers office, did it interfere with their working duties, was it scheduled at the beginning, the middle or the end of their working day? Did the participants perform their routine computer work? Were the computer tasks compatible with interruptions every 10 minutes? | The study was conducted in a research laboratory at a University. The laboratory contained a private room with a sit-stand workstation. All participants were University employees and they undertook their usual computer work during the study. The study was scheduled to accommodate the participant’s work schedule.
The following text has been added to describe where the study was conducted: “The 1-hour testing sessions were conducted in a private room in a laboratory at the participants’ workplace. »
The following text in the methods section explains that the particpants completed their usual computer work during the study : “Participants completed a prolonged standing task [15] while performing their usual computer-based work duties for an hour under two different conditions….”
|
4 | If conducted at the university, how long were participants off duty on the date of the experiment? All this information is essential for understanding, whether the use of a standing workstation is a valid alternative in the real working life of patients with mild back pain. | Participants performed their regular computer work during the experimental session. Thus, participants were not “off duty” to attend the experiment. Further, as the testing occurred at in a laboratory at the participants’ workplace, interruption to attend testing was minimal.
This has been address with the added text described in the comment above.
|
5 | Student´s t-test requires a normal distribution of data. Was this checked? | Thank you for this comment and drawing our attention to this. Data were not normally distributed and as such Wilcoxon Paired Signed Rank tests were used instead of Paired T-tests and data are presented as median and inter-quartile range instead of mean and standard deviation.
The following text has been added to the methods: “As data were not normally distributed, Wilcoxon Paired Signed Rank tests were used to compare changes in pain severity between conditions and data are presented as median and inter-quartile range (IQR).”
The presentation of data and= p-values have been changed in the results to represent this change in statistical analyses and presentation of data.
|
6 | Whilst pain worsening is clearly defined as increase of NRS score equal or greater than 2 points, a definition of improvement (= pain reduction?) is missing. | The text in the methods has been modified to indicate that participants who report their pain to be 2 points lower on the NRS at the end of the standing tasks is also reported.
Edited text: “The number (percentage) of participants who reported their pain to be 2 points higher or lower on the NRS at the end of the standing tasks (which suggests a clinically meaningful change in pain [20]) are reported.”
The following text has been added to the results to present data for the number of participants who reported an improvement in pain of ³2 points on the numerical rating scale.
Text added to the results: “Two out of 16 participants (13%) reported an improvement in pain of ³2 points on the NRS for each of the conditions.”
|
7 | The body mass index should be added to anthropometric characteristics of participants. | Body mass index has been added to the anthropometric characteristics of participants. |

Round 2
Reviewer 1 Report
All concerns have been adequately addressed.
Author Response
Thank you for your feedback and assistance in improving this paper.
Reviewer 2 Report
The manuscript has improved, and all points criticised in the previous version have been addressed. The setup of the experiment is now clear, and the method of the statistical analysis appropriate. The course of pain and their variation between individuals is better described.
However, the exchange of mean and standard deviation with median and interquartile range is incompletely performed, upper and lower quartile should be separated by a semicolon. The text in lines 148-152, page 4 must be corrected accordingly “There was no significant difference in the change in maximal severity of LBP pain experienced throughout the prolonged standing task between the footstool (mean median (SDIQR): 1.3 (1.71.5)) and no footstool (median (IQR): mean (SD): 1.3 (1.72)) conditions (p=0.8068). The change in the severity of LBP experienced by participants from the start to the end of the 60-minutes standing task was also similar between conditions (median (IQR) footstool: 0 .1 (2.10.75),; no footstool: 0.81 (2.32); p=0.2614).
Missing punctuation marks (line 37), word spacing (lines 38, 52, 58) and spelling (upper case after comma, line 41; “should read “compare” instead of “compared”, line 64) must be corrected. Some extra words remaining after amending the first version such as “using paired t-tests” page 3, line121, or “Standing using a” page 5, line122, must finally be deleted.
On page 5, lines 213-214, the sentence “Moving the foot onto and off of a footstool would be expected to mayfacilitate increased movement and having the foot position on or off of the footstool would may add variability to sagittal plane posture” requires rephrasing.
Amendments of the mentioned points and another check of the revised manuscript is recommended before publication may be considered.
Author Response
Dr Miya Zhang
Assistant Editor
IJERPH
12 April 2019
Dear Dr Zhang,
Re: Manuscript ijerph-478526: The effect of using a footstool during a prolonged standing task on low back pain in office workers
Thank you very much for giving us the opportunity to revise our submitted manuscript. We have addressed the comments from Reviewer #2 on a point by point basis in the table below and made amendments to the manuscript accordingly.
Thank you again and looking forward to hearing from you in due course.
Yours sincerely,
Michelle Smith
Response to comments from Reviewers
Reviewer’s comment | Author response | |
1 | The manuscript has improved, and all points criticised in the previous version have been addressed. The setup of the experiment is now clear, and the method of the statistical analysis appropriate. The course of pain and their variation between individuals is better described. | Thank you for your feedback and assistance in improving this paper. |
3 | However, the exchange of mean and standard deviation with median and interquartile range is incompletely performed, upper and lower quartile should be separated by a semicolon. The text in lines 148-152, page 4 must be corrected accordingly “There was no significant difference in the change in maximal severity of LBP pain experienced throughout the prolonged standing task between the footstool (mean median (SDIQR): 1.3 (1.71.5)) and no footstool (median (IQR): mean (SD): 1.3 (1.72)) conditions (p=0.8068). The change in the severity of LBP experienced by participants from the start to the end of the 60-minutes standing task was also similar between conditions (median (IQR) footstool: 0 .1 (2.10.75),; no footstool: 0.81 (2.32); p=0.2614). | The presentation of inter-quartile range (IQR) has been changed to present the upper and lower (25% and 75%) quartiles, rather than the range between these quartiles, as suggested by the reviewer.
Previous text: “There was no significant difference in the change in maximal severity of LBP pain experienced throughout the prolonged standing task between the footstool (median (IQR): 1 (1.5)) and no footstool (median (IQR): 1 (2)) conditions (p=0.68). The change in the severity of LBP experienced by participants from the start to the end of the 60-minutes standing task was also similar between conditions (median (IQR) footstool: 0 (0.75), no footstool: 1 (2); p=0.14).”
Revised text: “There was no significant difference in the change in maximal severity of LBP pain experienced throughout the prolonged standing task between the footstool (median (IQR): 1 (0; 1.25)) and no footstool (median (IQR): 1 (0; 2)) conditions (p=0.68). The change in the severity of LBP experienced by participants from the start to the end of the 60-minutes standing task was also similar between conditions (median (IQR) footstool: 0 (-0.13; 0.25), no footstool: 1 (0; 2); p=0.14).”
|
4 | Missing punctuation marks (line 37), word spacing (lines 38, 52, 58) and spelling (upper case after comma, line 41; “should read “compare” instead of “compared”, line 64) must be corrected. Some extra words remaining after amending the first version such as “using paired t-tests” page 3, line121, or “Standing using a” page 5, line122, must finally be deleted.
| These corrections have been made where possible in the paper. The extra words from the previous revision were not observed to be present. |
5 | On page 5, lines 213-214, the sentence “Moving the foot onto and off of a footstool would be expected to mayfacilitate increased movement and having the foot position on or off of the footstool would may add variability to sagittal plane posture” requires rephrasing.
| These corrections have been made. |
6 | ||